# Tree diversity in agroforestry systems of native fine-aroma cacao, Amazonas, Peru

**Malluri Goñas** [1]*, **Karol B. Rubio**[1], **Nilton B. Rojas Briceño**[1], **Elí Pariente-Mondragón**[2], **Manuel Oliva-Cruz**[1]

1 Instituto de Investigación para el Desarrollo Sustentable de Ceja de Selva, Universidad Nacional Toribio Rodríguez de Mendoza de Amazonas, Chachapoyas, Perú, 2 Facultad de Ingeniería y Ciencias Agrarias, Universidad Nacional Toribio Rodríguez de Mendoza de Amazonas, Chachapoyas, Perú

* malluri.gonas@untrm.edu.pe

## Abstract

Cocoa cultivation is of considerable economic and social importance to the Amazonas region and is commonly associated with forest species in the region. However, the diversification level and composition of cacao agroforestry systems in Peru are poorly understood. The objective of this study is, therefore, to describe the diversity of tree species in cocoa AFS by plantation age. Accordingly, the number of species of 15 plots covering a total of 1.5 hectares was recorded. Moderately low levels of tree species diversity were reported (H´ ranged 0.89–1.45). In total 17 species were reported throughout the study area. The most abundant botanical family was represented by a single *Musa sp.* species. The dissimilarity indices show a moderate similarity between the age ranges evaluated (over 62%). Additionally, the IVI indicates that the most important species are used for food and timber apart from providing shade, additionally major of this species are introduced intentionally for the farmers. Based on the observations, it may be concluded that the farmer's interest in obtaining further benefits from the plot, mostly economic benefits affect the diversification of cocoa agroforestry systems.

**Data Availability Statement:** All relevant data are within the paper and its Supporting information files.

**Funding:** The authors thank the Fondo Nacional de Desarrollo Científico, Tecnológico y de Innovación

## Introduction

Cocoa (*Theobroma cacao* L.) is one of the main crops grown in agroforestry systems (AFS) by smallholder farmers in humid tropics [1]. The installation of cocoa farms occurs mainly by partially clearing forests; the retained trees provide shade for cocoa and coproduction channels to farmers, while leaf mulch from shade trees and nutrients stored in the forest soil ensure productivity [2, 3].

Cocoa AFS is a socio-economically viable system in which farmers intentionally integrate shade trees, cocoa, and food crops together on the same plot [4]. Organic cocoa AFS uses a variety of shade trees to both suppress weed growth and insect pests outbreaks [5, 6] and compensate for nutrient loss due to nutrient uptake by cocoa trees through nitrogen fixation, reduction of nutrient leakage and decomposition of shade tree litter [6].

Cocoa AFS conserve the diversity of native plants and animals [7] and provide products that diversify farmers' diets. It generates supplemental income and some security from climate

Tecnológica (FONDECYT) for funding this research through the Contract N° 026-2016 of the "Círculo de Investigación para la Innovación y el fortalecimiento de la cadena de valor del cacao nativo fino de aroma en la zona nor oriental del Perú-CINCACAO" project, executed by the Instituto de Investigación para el Desarrollo Sustentable de Ceja de Selva (INDES – CES), de la Universidad Nacional Toribio Rodríguez de Mendoza de Amazonas. The funders had no role in study design, data collection and analysis, decision to publish, or preparation of the manuscript.

**Competing interests:** The authors have declared that no competing interests exist.

change-related shocks [1, 8, 9]. Moreover, cocoa trees benefit from an improved microclimate and increased water retention [1, 5, 10]. Therefore, cocoa cultivation in AFS is an alternative to contribute significantly to the mitigation of biodiversity loss and ecosystem resilience in tropics regions [9, 11], especially within areas where forest cover has been significantly reduced [12]. Furthermore, it has been shown that an integration of organic agriculture and agroforestry would effectively enhance biodiversity conservation [9, 13].

However, there is a growing global demand; and despite the current challenging economic and geopolitical world situation in the first half of the 2021/2022 season, the cacao demand sustained a positive stance and the prices also sustained this hike [14]. This would lead to farmers putting even more pressure on cocoa-producing ecosystems, which along with the scarcity of information to support the above claims and the importance of tree diversity in cocoa agroecosystems would worsen conservation and recovery of disturbed areas.

In the Amazonas region, there are no studies yet that show the diversity of trees associated with cocoa cultivation; therefore, the present work aims to identify the species present within the AFS and measure the Diameter at Breast Height (DBH). These data allowed to obtain: 1) The diversity, abundance and importance value of shade trees within the system; 2) The composition and spatial distribution of trees in the AFS of cocoa in the Amazon.

## Materials and methods

### Study area

The study was carried out in cocoa AFS belonging to members of the APROCAM Multiple Service Cooperative. The APROCAM cooperative consists of 235 small producers distributed in 4 districts in the province of Bagua (Aramango, Copallín, La Peca and Imaza), 2 districts in the province of Utcubamba (Cajaruro and El Parco) and 1 district in the province of Santa María de Nieva (Nieva) in the Amazonas. The climate in the area is warm tropical with an average temperature of 25˚C and annual rainfall between 500–1500 mm.

### Data collection methods

In the aforementioned territories, 15 plots were selected (Fig 1) according to the age of the crop. The plots were grouped into three strata (young cocoa AFS: 8–15 years; middle-aged cocoa AFS: 16–29 years and old cocoa AFS: 30–40 years); these three age groups were determined because cocoa reaches its peak production at 8 years of age and stabilizes until 15 years of age, from 16 to 30 years of age cocoa production is maintained on average and finally after 30 years of age the yield declines modernly and becomes an adult cocoa with a low yield [16, 17]. In addition, each of the farms selected were at least 1.5 ha in size.

Data collection was carried out from November to December 2020. In each selected plot, a 50 m x 20 m rectangular subplot (sample) was established and the number of species within this subplot was recorded. In total, an area of 1.5 ha of cocoa cultivated area was sampled.

Tree species were identified at the field itself, since the trees found (fruit and timber species) were widely known by technicians. On the contrary, for species where it was not possible to taxonomically identify, only the common name provided by farmers and a botanical sample were taken for later identification. This botanical identification was carried out with the help of specialized bibliography and expert consultation in the KUELAP-UNTRM herbarium (Campus Universitario: C. Universitaria N˚ 304, Chachapoyas, Amazonas, Perú). Additionally, the recognized species were classified as native or exotic according to their origin; and as forest, medicinal or timber species according to their main use by farmers or potential products.

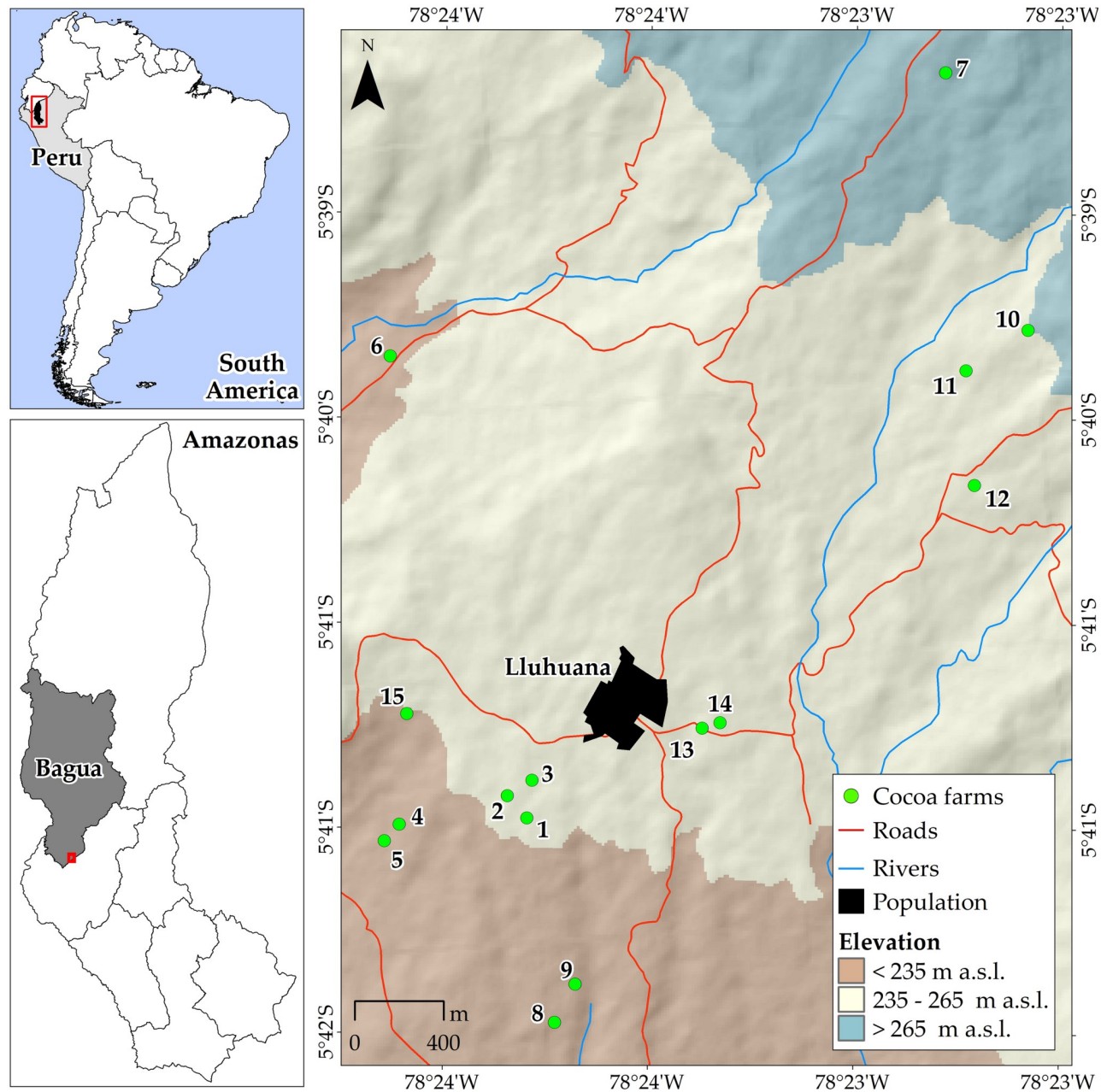

**Fig 1. Plot location map.** Map prepared by the authors based on open access resources: political-administrative boundaries from geoBoundaries [15], local details of ARA-Amazonas (http://visor.regionamazonas.gob.pe/indexv.php) and ALOS PALSAR digital elevation model (https://search.asf.alaska.edu/#/), 28 of June 2022.

The Circumference at Breast Height (CBH) of each of the stems of the tree species including cocoa trees was taken at 1.3 m above the ground. This data was transformed into Diameter at Breast Height (DBH) of each of the measured stems. For this transformation we used "π" as a circumference division factor.

## Data processing and analysis

**Diversity of shade trees.** The data obtained were processed with PAST 3 software to generate a diversity analysis. For each plantation and age range, species richness (S) and the

effective number of species (N) indices were calculated, as well as Shannon (H'), Simpson (1-D), Margalef (Dmg) [18], and Chao-1 indices [19]. Excel spreadsheets were used to manage tables and graphs, such as the Venn diagram for species richness.

The Importance Value Index was estimated using the following formula developed by Curtis and McIntosh [20]:

$$IVI\,(\%) : RA(\%) + RF(\%) + RD(\%) \tag{1}$$

Where: RA: is relative abundance calculated as the number of individuals per species. RF: is relative frequency estimated as the proportion of plots where the species occurred at least once. RD: is relative dominance defined as the basal area per species per hectare.

**Composition and spatial distribution of trees in the cocoa agroforestry system.** To evaluate the species composition in each age range, the Jaccard and Sorensen indices were used [18]. In addition, the trees were grouped according to their potentially usable products (wood, fruit, or medicinal). The basal area of all shade species and cocoa trees was also established.

**Statistical analysis.** All data that conformed to ANOVA assumptions were processed using this analysis to establish differences between age groups. All analyses were performed using Infostat v.2020 software and at a significance level of 5% (p<0.05) [21].

## Results

### Diversity, abundance, and importance value

A total of 454 individuals distributed in 14 families were inventoried in this study (Table 1). Only two families exceeded 50 individuals being the most abundant (Musaceae and Boraginaceae). In general, when visualizing the merged data for species per botanical family (Table 1), the families with the highest species richness are Anacardiaceae (2 species), Fabaceae (2 species), and Rutaceae (4 species). It is also observed that the highest number of shade tree individuals was inventoried in the youngest plots (8–15 years old) with 206 individuals.

Species richness was the lowest at middle aged (16–29 years) cacao plots (Table 2). In fact, there were a total of 17 species observed in the study. The estimated total number of species based on Chao-1 was 15, 9.5 and 15 for young cocoa AFS, middle-age cocoa AFS, and old cocoa AFS respectively. Simpson's (1-D) and Shannon's (H) indices ranged from 0.34 to 0.63

**Table 1. Number of species and abundance of individuals by botanical family in each age range of AFS and as a total.**

| | Young | | Middle-aged | | Old | | Total | |
|---|---|---|---|---|---|---|---|---|
| Familia | Spec. | Indiv. | Spec. | Indiv. | Spec. | Indiv. | Spec. | Indiv. |
| Anacardiaceae | 2 | 2 | 1 | 3 | 1 | 2 | 2 | 7 |
| Annonaceae | 1 | 2 | 0 | 0 | 0 | 0 | 1 | 2 |
| Arecaceae | 1 | 1 | 1 | 3 | 0 | 0 | 1 | 4 |
| Malvaceae | 0 | 0 | 0 | 0 | 1 | 2 | 1 | 2 |
| Boraginaceae | 1 | 14 | 1 | 29 | 1 | 12 | 1 | 55 |
| Caricaceae | 0 | 0 | 0 | 0 | 1 | 5 | 1 | 5 |
| Fabaceae | 2 | 7 | 1 | 8 | 2 | 8 | 2 | 23 |
| Lauraceae | 1 | 2 | 1 | 5 | 1 | 5 | 1 | 12 |
| Meliaceae | 1 | 3 | 1 | 1 | 1 | 3 | 1 | 7 |
| Musaceae | 1 | 167 | 1 | 63 | 1 | 86 | 1 | 316 |
| Rubiaceae | 1 | 3 | 0 | 0 | 1 | 2 | 1 | 5 |
| Rutaceae | 3 | 5 | 2 | 3 | 3 | 8 | 4 | 16 |
| Total | 14 | 206 | 9 | 115 | 13 | 133 | 17 | 454 |

**Table 2. Diversity indices by the altitudinal range and in general.**

|  | Young | Middle-aged | Old | Total |
|---|---|---|---|---|
| Species Richness (*S*) | 14 | 9 | 13 | 17 |
| Individuals | 206 | 115 | 133 | 454 |
| Simpson_1-D | 0.34 | 0.63 | 0.57 | 0.4975 |
| Shannon_H | 0.89 | 1.34 | 1.45 | 1.271 |
| Margalef | 2.44 | 1.69 | 2.45 | 2.615 |
| Chao-1 | 15.00 | 9.50 | 13.00 | 17.33 |

and 0.89 to 1.45 respectively. In general, in the study area, middle-age AFS are the least diverse compared to young and old AFS (Table 2).

Thus, the 454 individuals were distributed among 17 species, of which 6 are native and 11 are exotic or introduced individuals. The species with the highest abundance were *Musa sp* (316 individuals in total), *Cordia alliodora* (55 individuals in total), and *Inga spp* (16 individuals) while *Spondias purpurea* and *Citrus aurantifolia* are the least abundant with only 1 individual recorded in the whole study area.

The IVI reveals that in the three age ranges of the AFS the two most important species are *Musa sp* and *Cordia alliodora* with a total IVI to 115.31% and 87.06% respectively (Table 3). In the young AFS plots, there are only 4 species that exceed 10% IVI. While in the middle-age AFS plots, there are up to 9 species exceeding this limit.

Fruit and timber species such as *Musa sp* and *Cordia alliodora* are the most dominant in the evaluated plots, and all species cultivated for cocoa shade have other potential uses, whether fruit, timber, or medicinal (Table 4). In all the evaluated plots, there is a majority of species with a low density of individuals.

## Composition and spatial distribution of trees in cacao AFS

The analysis of variance does not show significant statistical differences between the basal area of shade trees, basal area of cocoa trees, and total basal area ($p < 0.005$). However, it can be observed that as the cacao plantation gets older the basal area of cocoa trees increases. Total basal area under young, middle-aged and old cacao AFS was observed to be 8.23 m$^2$ ha$^{-1}$, 9.96 m$^2$ ha$^{-1}$ and 10.95 m$^2$ ha$^{-1}$ respectively. Likewise, shade trees occupied a greater average basal area (15.92 m$^2$ ha$^{-1}$) in middle-aged cocoa AFS, while less in old cocoa AFS (4.39 m$^2$ ha$^{-1}$) and young cocoa AFS reached 5.6 m$^2$ ha$^{-1}$ of basal area for shade trees (Fig 2). Similarly, the total basal area was greater in middle-aged cocoa AFS than in young or old cacao AFS. Only in middle-aged cocoa AFS was the basal area occupied by shade trees (61.51%) greater than the area occupied by cocoa trees (38.49%). In young and adult cocoa AFS, cocoa occupied a basal area of 59.48% and 71.36% respectively of the total basal area.

Seven of the total species studied were observed to be common in all the three age categories (Table 2; Fig 3). *Spondias purpurea* and *Annona muricata* were only reported in young cacao AFS, while *Matisia cordata*, *Carica papaya* and *Citrus aurantifolia* were only observed in old cacao AFS. Dissimilarity indices are higher when comparing young cocoa AFS with middle-aged or old cocoa AFS than when comparing old cocoa AFS with middle-aged cocoa AFS (Fig 4).

## Discussion

### Diversity, abundance, and importance value

In our study, AFS cocoa can be differentiated among three age groups based on their diversity, species abundance and predominant species. The two predominant species in the three age

**Table 3. Most important species (IVI>10) by plantation age and total.**

|  | RA[a] (%) | RF[b] (%) | RD[c] (%) | IVI[d] (%) |
|---|---|---|---|---|
| **Young** |  |  |  |  |
| *Musa sp* | 81.07 | 14.29 | 69.85 | 165.20 |
| *Cordia alliodora* | 6.80 | 19.05 | 14.68 | 40.53 |
| *Lysiloma divaricatum* | 0.97 | 9.52 | 2.82 | 13.32 |
| *Calycophyllum spruceanum* | 1.46 | 9.52 | 0.21 | 11.19 |
| **Middle-aged** |  |  |  |  |
| *Cordia alliodora* | 25.22 | 22.22 | 83.51 | 130.95 |
| *Musa sp* | 54.78 | 22.22 | 9.39 | 86.40 |
| *Inga sp.* | 6.96 | 11.11 | 1.92 | 19.99 |
| *Cocus nucifera L.* | 2.61 | 11.11 | 1.66 | 15.38 |
| *Citrus sinensis* | 1.74 | 11.11 | 0.04 | 12.89 |
| *Persea Americana* | 4.35 | 5.56 | 0.23 | 10.14 |
| *Old* |  |  |  |  |
| *Musa sp* | 64.66 | 18.18 | 40.12 | 122.96 |
| *Cordia alliodora* | 9.02 | 4.55 | 33.72 | 47.28 |
| *Persea Americana* | 3.76 | 13.64 | 4.75 | 22.15 |
| *Lysiloma divaricatum* | 3.76 | 9.09 | 6.61 | 19.46 |
| *Carica papaya* | 3.76 | 9.09 | 0.75 | 13.60 |
| *Cedrela odorata* | 2.26 | 9.09 | 1.47 | 12.82 |
| *Citrus limetta* | 1.50 | 9.09 | 0.38 | 10.98 |
| *Matisia cordata* | 1.50 | 4.55 | 4.90 | 10.95 |
| *Citrus reticulata* | 3.76 | 4.55 | 2.37 | 10.67 |
| *Total* |  |  |  |  |
| *Musa sp* | 69.60 | 18.03 | 27.67 | 115.31 |
| *Cordia alliodora* | 12.11 | 14.75 | 60.19 | 87.06 |
| *Persea Americana* | 2.64 | 8.20 | 1.31 | 12.15 |
| *Inga sp.* | 3.52 | 6.56 | 1.89 | 11.97 |

[a] RA = Relative abundance.

[b] RF = Relative frequency.

[c] RD = Relative dominance.

[d] IVI = Importance Value Index.

levels of the cocoa AFS are *Musa sp* and *Cordia alliodora*, the first cultivated mainly for its highly marketable fruit and the second a timber specie with an important local market. In general, all the species reported are exploited either for the sale or consumption of their fruits or wood. Similarly, in cocoa AFS in Ivory Coast, Tondoh et al. [2] reported the predominance of fruit tree species in AFS and indicated that farmers planted fruit trees to provide shade and income.

The diversity reported in our study is in line with that reported in other latitudes for cacao AFS. In Central America and Mexico, Shannon's diversity index (H′) and species richness (S) ranged from 0.8–2.99 and 22–104 [22, 23]. In the Amazon and Atlantic Forest biomes, there are AFS with variation in the number of families (11–45), S (13–180), and H′ (1.37–2.73) [24–26].

Most species found in the study have alternative uses such as food and timber. This diversification promotes the sustainability of cocoa AFS since it generates greater economic security for farmers [5, 27, 28]. This means that AFS generates social and environmental benefits

**Table 4. Species and abundance according to the age of AFS and in a total.** And the potential products that can be exploited from them. (E = Exotic; N = Native; F = Forest; M = Medicinal; T = Timber).

| Specie | Family | Young | Middle-aged | Old | Total | Native[a] | Potential Products[b] |
|---|---|---|---|---|---|---|---|
| *Mangifera indica* | Anacardiaceae | 1 | 3 | 2 | 6 | E | F |
| *Spondias purpurea* | Anacardiaceae | 1 | 0 | 0 | 1 | E | F |
| *Annona muricata* | Annonaceae | 2 | 0 | 0 | 2 | N | F, M |
| *Cocus nucifera L.* | Arecaceae | 1 | 3 | 0 | 4 | N | F |
| *Matisia cordata* | Malvaceae | 0 | 0 | 2 | 2 | N | F |
| *Cordia alliodora* | Boraginaceae | 14 | 29 | 12 | 55 | E | T |
| *Carica papaya* | Caricaceae | 0 | 0 | 5 | 5 | E | F |
| *Inga sp.* | Fabaceae | 5 | 8 | 3 | 16 | N | F |
| *Lysiloma divaricatum* | Fabaceae | 2 | 0 | 5 | 7 | N | T |
| *Persea americana* | Lauraceae | 2 | 5 | 5 | 12 | E | F |
| *Cedrela odorata* | Meliaceae | 3 | 1 | 3 | 7 | E | T |
| *Musa sp* | Musaceae | 167 | 63 | 86 | 316 | E | F |
| *Calycophyllum spruceanum* | Rubiaceae | 3 | 0 | 2 | 5 | N | T |
| *Citrus limetta* | Rutaceae | 2 | 1 | 2 | 5 | E | F |
| *Citrus sinensis* | Rutaceae | 1 | 2 | 0 | 3 | E | F |
| *Citrus reticulata* | Rutaceae | 2 | 0 | 5 | 7 | E | F |
| *Citrus aurantifolia* | Rutaceae | 0 | 0 | 1 | 1 | E | F |

[a] It is indicated the species is native (N) or exotic (E).

[b] It is indicated the potential products that can be exploited from each specie (F = Forest; M = Medicinal; T = Timber).

without affecting the economic aspect of agricultural production [29–31]. Further, the cultivation of cocoa in AFS improves the conservation of floral diversity and management practices [23].

This trend toward the dominance of fruit, timber, and medicinal trees could be a strong indication of the deliberate transformation of the landscape by farmers from natural pioneer species that have traditionally been grown with cocoa to species that provide food and medicinal benefits [4]. Furthermore, considering that the agroecosystems evaluated in our study are spaces with a high level of anthropization, it explains the relatively low levels of diversity and the predominance of species that in addition to shade provide other benefits to the producer like wood and fruit harvest and traditional uses as medicinal trees.

Likewise, well-planned shade trees influences positive impact on cacao yield, improves net income together with various ecosystem services subject to implementation of proper management strategies [6, 32–35]. In addition, diversifying the agroecosystem with shade trees improves overall agroecosystem performance as each of these species fulfill one or more essential functions in nutrient cycling [36, 37], pest and disease control [32, 38], biomass production and carbon stocks [10, 37, 39–41].

## Composition and spatial distribution of trees in cocoa AFS

Micro-parceling was also a common phenomenon observed in the study area because of the presence relatively small cultivable plots (0.5–2 ha). Asare et al. [42] and Asigbaase et al. [4] also observed micro-parceling in other places where cocoa is produced. In terms of basal area, although no significant differences are observed, higher values of the basal area occupied by cocoa trees are observed in middle-aged or old cocoa AFS rather than in young AFS. This is probably because, in older AFS, cocoa trees have been able to develop a greater stem thickness.

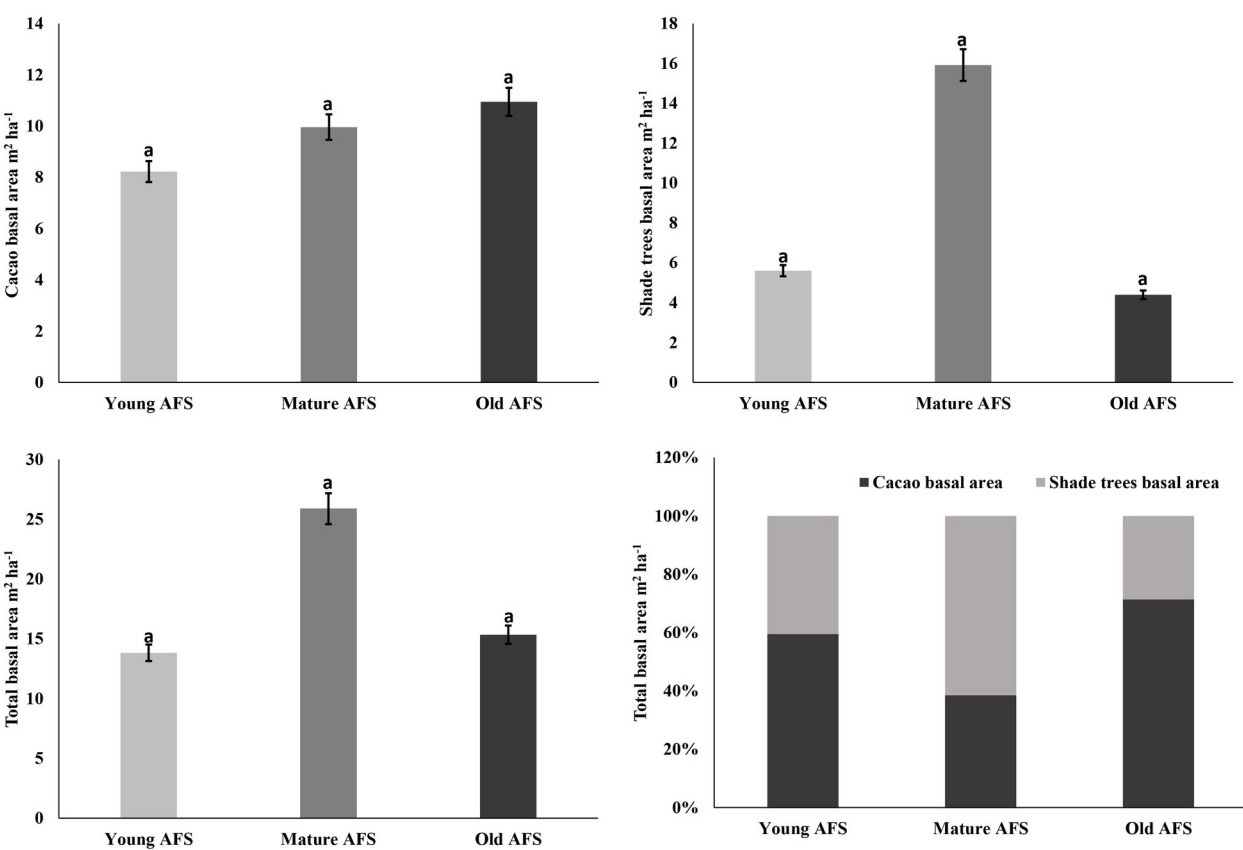

**Fig 2. Basal area of cocoa, shade trees, in a total and basal area of cocoa and shade trees as a percentage of the total.** Means with equal letters are not statistically different (p<0.05).

However, when comparing the average basal area of shade trees, it is observed that middle-aged cocoa AFS occupy a larger basal area despite having a lower density of total shade trees. This is probably due to the presence of many mature timber trees such as *Cordia alliodora*. These phenomena are also reported by Asigbaase et al. [4] when they evaluated organic and conventional cocoa AFS in Ghana.

Jaccard and Sorensen's dissimilarity indices are above 50% when comparing the age groups with each other, evidencing moderate similarity between these groups. Other authors such as Asigbaase et al. [4] and Braga et al. [43] report high levels of dissimilarity; however, the main difference is that Asigbaase et al. [4] made comparisons between organic and conventional cocoa trees while in this study only organic cocoa trees are incorporated.

## Conclusions

This study contributes to the knowledge of the composition and diversity of tree species communities in cocoa agroecosystems in the Amazonas region. Although no marked differences in diversity are observed and the dominance of two species used as shade is evident, it contributes to the understanding of how cocoa farmers modify and diversify the PBS.

The high levels of similarity observed are combined with the preference of cocoa farmers to grow timber or fruit tree species alongside cocoa, mostly native, diversifying their plots and obtaining economic and consumption benefits in addition to cocoa farming. This

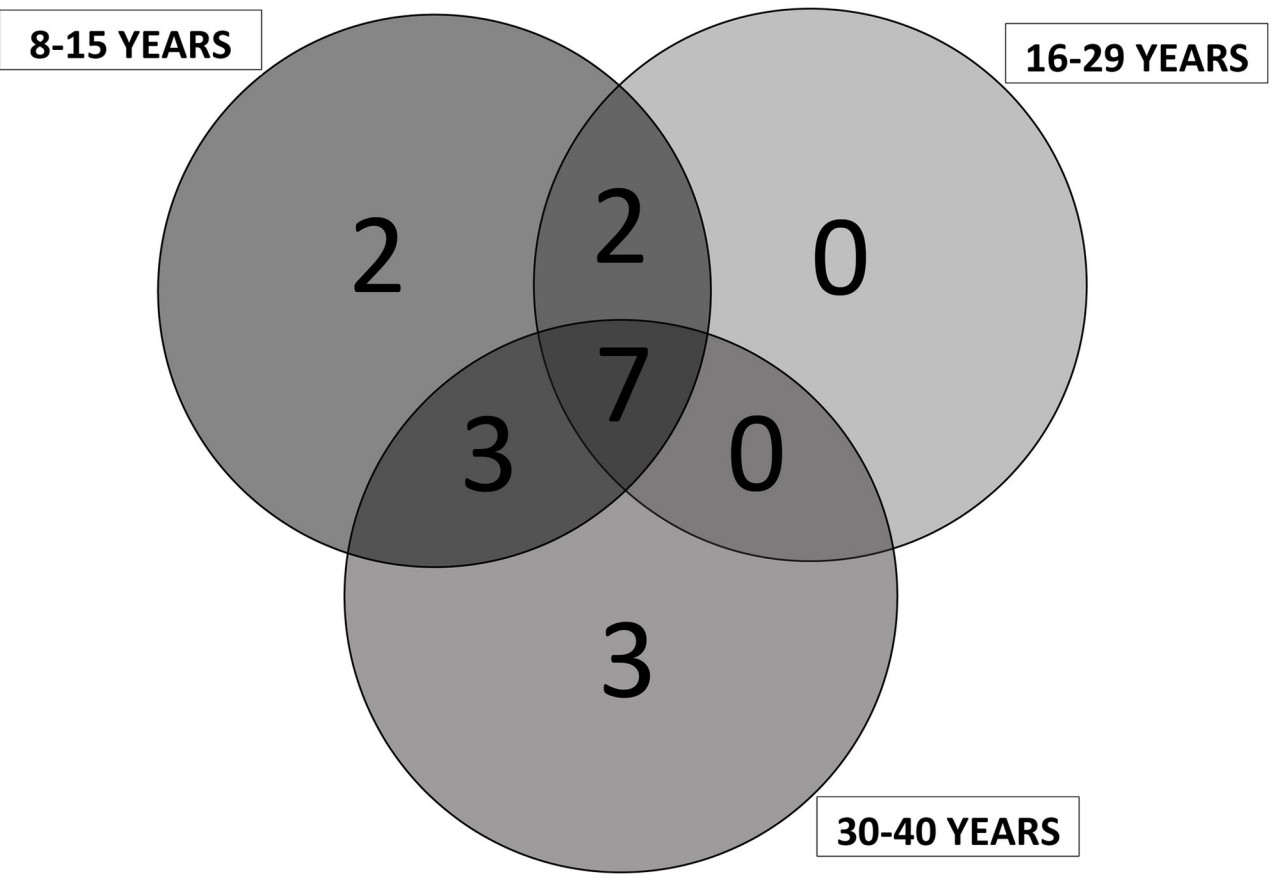

**Fig 3. Number of species per age level of cocoa AFS.**

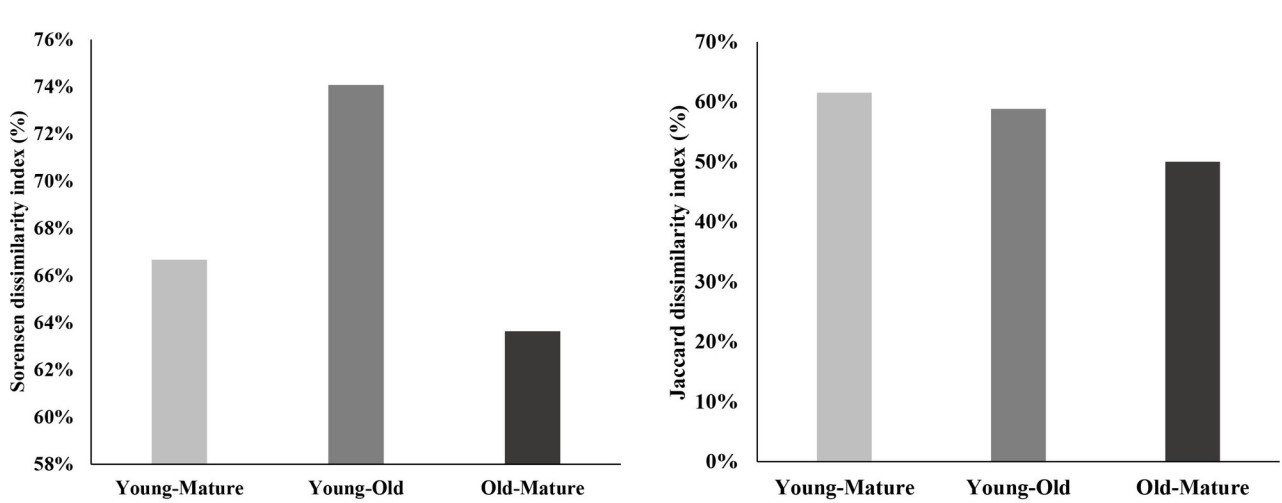

**Fig 4. Species dissimilarity between cocoa plots according to age groups.**

diversification, however, should be managed to improve the distribution of shade trees and improve diversification management practices that are beneficial to farmers and the environment.

## Supporting information

**S1 File. Data base.**
(XLSX)

## Author Contributions

**Conceptualization:** Malluri Goñas, Nilton B. Rojas Briceño, Elí Pariente-Mondragón, Manuel Oliva-Cruz.

**Data curation:** Malluri Goñas.

**Formal analysis:** Malluri Goñas, Karol B. Rubio, Nilton B. Rojas Briceño, Manuel Oliva-Cruz.

**Investigation:** Malluri Goñas.

**Methodology:** Malluri Goñas, Karol B. Rubio, Elí Pariente-Mondragón.

**Resources:** Manuel Oliva-Cruz.

**Writing – original draft:** Malluri Goñas, Karol B. Rubio, Nilton B. Rojas Briceño, Elí Pariente-Mondragón.

**Writing – review & editing:** Malluri Goñas, Karol B. Rubio, Nilton B. Rojas Briceño, Manuel Oliva-Cruz.

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
