## [Decision Letter · Decision Letter 0]

17 May 2022

PONE-D-22-03463Tree diversity in agroforestry systems of native fine-aroma native cacao, Amazonas, PeruPLOS ONE

Dear Dr. Goñas,

Thank you for submitting your manuscript to PLOS ONE. After careful consideration, we feel that it has merit but does not fully meet PLOS ONE’s publication criteria as it currently stands. Therefore, we invite you to submit a revised version of the manuscript that addresses the points raised during the review process.

ACADEMIC EDITOR: The reviewer has provided insightful comments. Addressing those adequately will enhance the quality of the manuscript. I request authors to undertake a thorough revision considering all the comments of the reviewer.

We look forward to receiving your revised manuscript.

Kind regards,

Arun Jyoti Nath

Academic Editor

PLOS ONE

Journal Requirements:

“The authors thank the Fondo Nacional de Desarrollo Científico, Tecnológico y de Innovación Tecnológica (FONDECYT) for funding this research through the Contract N° 026-2016 of the "Círculo de Investigación para la Innovación y el fortalecimiento de la cadena de valor del cacao nativo fino de aroma en la zona nor oriental del Perú-CINCACAO" project, executed by the Instituto de Investigación para el Desarrollo Sustentable de Ceja de Selva (INDES – CES), de la Universidad Nacional Toribio Rodríguez de Mendoza de Amazonas.  “

“The authors thank the Fondo Nacional de Desarrollo Científico, Tecnológico y de Innovación Tecnológica (FONDECYT) for funding this research through the Contract N° 026-2016 of the "Círculo de Investigación para la Innovación y el fortalecimiento de la cadena de valor del cacao nativo fino de aroma en la zona nor oriental del Perú-CINCACAO" project, executed by the Instituto de Investigación para el Desarrollo Sustentable de Ceja de Selva (INDES – CES), de la Universidad Nacional Toribio Rodríguez de Mendoza de Amazonas.”

“The authors thank the Fondo Nacional de Desarrollo Científico, Tecnológico y de Innovación Tecnológica (FONDECYT) for funding this research through the Contract N° 026-2016 of the "Círculo de Investigación para la Innovación y el fortalecimiento de la cadena de valor del cacao nativo fino de aroma en la zona nor oriental del Perú-CINCACAO" project, executed by the Instituto de Investigación para el Desarrollo Sustentable de Ceja de Selva (INDES – CES), de la Universidad Nacional Toribio Rodríguez de Mendoza de Amazonas.  “

7. We note that Figure 1 in your submission contain map images which may be copyrighted. All PLOS content is published under the Creative Commons Attribution License (CC BY 4.0), which means that the manuscript, images, and Supporting Information files will be freely available online, and any third party is permitted to access, download, copy, distribute, and use these materials in any way, even commercially, with proper attribution. For these reasons, we cannot publish previously copyrighted maps or satellite images created using proprietary data, such as Google software (Google Maps, Street View, and Earth). For more information, see our copyright guidelines: http://journals.plos.org/plosone/s/licenses-and-copyright.

Reviewers' comments:

Reviewer's Responses to Questions

**Comments to the Author**

1. Is the manuscript technically sound, and do the data support the conclusions?

Reviewer #1: Partly

2. Has the statistical analysis been performed appropriately and rigorously? 

Reviewer #1: Yes

3. Have the authors made all data underlying the findings in their manuscript fully available?

Reviewer #1: Yes

4. Is the manuscript presented in an intelligible fashion and written in standard English?

Reviewer #1: No

5. Review Comments to the Author

Reviewer #1: The manuscript at present form requires major modification in texts and Result presentation in tabular and or graphical form. Here are the comments that I find should be considered to increase the manuscript to a technically sound work:

1. Why there are two ‘native’ in title of the manuscript? Fine aroma is native or the Cacao is or both?

2. L13–15: Give a suitable research gap before pointing the objectives. The use of adverb in the sentence showed you have explained the research gap. However, ‘Cacao cultivation ..... species in the region’ could not speak the gaps in the knowledge base.

3. L17–18: use the value or value range within parenthesis for moderately low.

4. L20: put value/ value range for similarity within parentheses.

5. L20–22: Please rephrase this portion. It is not understandable that whether you are suggesting further diversification or concluding with present diversified systems at the first half of the sentence.

6. L23: While rephrasing, place this sentence separately before concluding statement such that it can leave a remark for conclusion. At the present form it showed an interpretation of result.

7. L24: I suggest instead of dissimilarity indices, you can use species dissimilarity. Why is ‘especies frutales’ mentioned in the keywords? I could not find same and similar word(s) anywhere in the whole manuscript except in the keywords.

8. L28–30: Whether cacao is the primary crop for which forests were partially cleared or it was for some other cultivations? Give a clearer statement in support of it.

9. L36: Give an appropriate reference to the term ‘conventional cacao system.’

10. L38: How all of a sudden it became a productive system? As just now you termed it to be simplified system requiring agrochemical dependency, which again was not properly managed by smallholders due to high input costs.

11. L39–43: This is good but, statement made- “this tendency towards ..... high input costs” and in this section are discrete.

12. L32–34: Is it reasonable to use the term agroforest/ agroforestry for Cacao system? Because, the system undergoes intense modification in respect to economic priority of the cultivars. Whereas, any agroforest should mimic the tree composition with the native forest or secondary forest. I think ‘Conventional Cacao System’ or ‘Conventional Cacao Agro-ecosystem’ would be more appropriate.

13. L56–57: Add more recent reference on expected global market demand and price hike on Cacao.

14. L63: Add specified objectives of the study.

15. L74–75: On what basis you are terming them as young, middle age, etc. Also elaborate why <8 years cacao cultivation was deselected?

16. L76: If it is 15×1.5 ha then result is 22.5 not 22.2.

17. It looks that your qudrats (sampling plots) covered an area of 1.5 ha and not 22.2/22.5 ha. Clearly state what was the total area under laid plot not the area that was covered under cultivation? Because, your sampling time i.e., November and December raised questions in covering so much big area under quadrat laying. The section ‘In the afore mentioned ... 22.2 ha sampled’ and ‘Data collection was ... sub-plot was recorded needed rigorous modification to make it clearer to the audience.

18. L84–85: Farmers are mostly aware of vernacular names not scientific name and hence modify the sentence as an appropriately.

19. L89: Give official identification address of ‘KUELAP-UNTRM’ within the parenthesis.

20. L91–92: If you write this way then give the division factor which was used in conversion of CBH to DBH.

21. L93: ‘Procesamiento y análisis de datos’ Change it to English.

22. L94–103: Give specific references to each of the analysis performed. For example for richness, for Shannon, Simpson, Margalef, Chao-1 indices, Jaccard and Sorenson.

23. L112: ‘In general, when visualizing the total data...’ Rephrase this portion.

24. L119: ‘Species richness ranged from 1-9 in the plots.’ Table 2 showed Species richness at different aged stands ranged between 9 and 17.

25. L119–120: Modify to – ‘Species richness was the lowest at middle aged (16–29 years) cacao plots.’ Are you observing only Cacao trees in the entire cultivation system? If not, then why you are mentioning middle aged cacao trees? It should be middle aged of the cultivation system.

26. L123–126: ‘Simpson's (1-D) and Shannon's .... middle-age AFS are the least diverse compared to young and old AFS.’ This should be in discussion. Cite results of the diversity indices here.

27. L128: At L121 you have mentioned that total 16 species were observed in the study area; here you are mentioning 454 individuals were distributed among 17 species distributed; and table 3 representing 17 species.

28. Table 3: Give references below table which was used to delineate the species as exotic, native etc. or add a section at method how you have distinguished this?

29. Table 3: What is potential products? Add a section in the manuscript how it was enumerated and how important it is in phytosociological studies?

30. L137–144: ‘The IVI reveals that .... with a low density of individuals.’ This is not proper result writing. You should give the Dr, Fr, Dor, IVI, etc. values representing the valuable species. This section is neither complete result nor complete discussion.

31. How you have calculated IVI? Add a section in the method. Because it looks absolute value, and you are giving percent symbol. Better to use RIVI, this will give more clear expression of observed species.

32. Figure 2: What are Joven, Madura and Adulta? If these are vernacular then give explanation of the term in figure caption.

33. L153–154: ‘there was a total of 8.23 m*ha2-1, 9.96 ..... and old cocoa AFS respectively.’ Should be written as- ‘Total basal area under young, middle-aged and old cacao AFS was observed to be 8.23 m2 ha-1, 9.96 m2 ha-1 and 10.95 m2 ha-1 respectively.

34. ‘m*ha2-1’: Correct it to ‘m2 ha-1’ across the whole manuscript. The digit 2 should be at superscript of m expressing as squared meter and -1 should be superscripted at ha denoting per hectare.

35. L169: ‘Of the total .... in the three age rang’ change it to – ‘Seven of the total species studied were observed to be common in all the three age categories.’

36. The term ‘reported’ should be changed to observed, found, etc.

37. Figure 4: Change the language of axis title to English.

38. Shannon diversity index (H0) is not the correct expression. It should be H´.

39. L185–191: ‘The Cacao ecological zone .... looks notably promising.’ This section is not relevant for your research. I suggest change these explanations with relevant texts.

40. At the very beginning of discussion avoid citing and linking with earlier reports. Start discussion with some productive phrases relevant to your findings.

41. L200–201: A well-planned shade trees influences positive impact on cacao yield, improved net income together with various ecosystem services subject to implementation of proper management strategies (References).

42. L213: ‘Côte d'Ivoire’, change it to English, if not possible give description within parenthesis.

43. L222: that additionally provides other benefits (name some other benefits in this parenthesis) together with shade.

44. L224–225: Micro-parcelling (give brief description of micro-parcelling) was also a common phenomena observed in the study area because of the presence relatively small cultivable plots (0.5–2 ha).

45. L230–231: ‘since the density ... higher density of cacao plants.’ This explanation is not required.

46. L235¬–236: ‘which in young AFS .... so they have a lower density.’ It is a obvious fact so, this portion is not required.

47. L239–240: Jaccard’s and Sorenson’s are similarity indices. They evaluate similarity not dissimilarity. You can evaluate inverse Jaccard or inverse Sorenson for dissimilarity estimation. In such instances you have to explain how the dissimilarity or inverse similarity was evaluated in the methodology.

6. PLOS authors have the option to publish the peer review history of their article (what does this mean?). If published, this will include your full peer review and any attached files.

Reviewer #1: **Yes: **Panna Chandra Nath

---

## [Author Response · Author response to Decision Letter 0]

3 Aug 2022

I. Journal Requirements:

We ensure that the manuscript meets with PLOS ONE´s requirements. 

The permissions for research and samples collection in agricultural lands is no required. 

We recheck it.

“The authors thank the Fondo Nacional de Desarrollo Científico, Tecnológico y de Innovación Tecnológica (FONDECYT) for funding this research through the Contract N° 026-2016 of the "Círculo de Investigación para la Innovación y el fortalecimiento de la cadena de valor del cacao nativo fino de aroma en la zona nor oriental del Perú-CINCACAO" project, executed by the Instituto de Investigación para el Desarrollo Sustentable de Ceja de Selva (INDES – CES), de la Universidad Nacional Toribio Rodríguez de Mendoza de Amazonas. “

Please state what role the funders took in the study. 

The authors confirm that: "The funders had no role in study design, data collection and analysis, decision to publish, or preparation of the manuscript."

“The authors thank the Fondo Nacional de Desarrollo Científico, Tecnológico y de Innovación Tecnológica (FONDECYT) for funding this research through the Contract N° 026-2016 of the "Círculo de Investigación para la Innovación y el fortalecimiento de la cadena de valor del cacao nativo fino de aroma en la zona nor oriental del Perú-CINCACAO" project, executed by the Instituto de Investigación para el Desarrollo Sustentable de Ceja de Selva (INDES – CES), de la Universidad Nacional Toribio Rodríguez de Mendoza de Amazonas.”

We delete this section.

7. We note that Figure 1 in your submission contain map images which may be copyrighted. All PLOS content is published under the Creative Commons Attribution License (CC BY 4.0), which means that the manuscript, images, and Supporting Information files will be freely available online, and any third party is permitted to access, download, copy, distribute, and use these materials in any way, even commercially, with proper attribution. For these reasons, we cannot publish previously copyrighted maps or satellite images created using proprietary data, such as Google software (Google Maps, Street View, and Earth). For more information, see our copyright guidelines: http://journals.plos.org/plosone/s/licenses-and-copyright.

We provide a information about map obtaining in the figure 1 caption. 

II. Review Comments to the Author

Below we described the actions that we make for response a reviewer recommendations/comment:

1. Why there are two ‘native’ in title of the manuscript? Fine aroma is native or the Cacao is or both?

There was a writing issue. It was corrected. 

2. L13–15: Give a suitable research gap before pointing the objectives. The use of adverb in the sentence showed you have explained the research gap. However, ‘Cacao cultivation ..... species in the region’ could not speak the gaps in the knowledge base.

14-15: A new sentence providing a research gap was included.

3. L17–18: use the value or value range within parenthesis for moderately low.

L18: A diversity index value range was provided. 

4. L20: put value/ value range for similarity within parentheses.

L21: Value was put.

5. L20–22: Please rephrase this portion. It is not understandable that whether you are suggesting further diversification or concluding with present diversified systems at the first half of the sentence.

L21-24: We were rephrasing this sentence. 

6. L23: While rephrasing, place this sentence separately before concluding statement such that it can leave a remark for conclusion. At the present form it showed an interpretation of result.

Done.

7. L24: I suggest instead of dissimilarity indices, you can use species dissimilarity. Why is ‘especies frutales’ mentioned in the keywords? I could not find same and similar word(s) anywhere in the whole manuscript except in the keywords.

L25: We take the recommendation, we use species dissimilarity instead dissimilarity indices. Also, instead “species frutales” we put the translation “fruit species”

8. L28–30: Whether cacao is the primary crop for which forests were partially cleared or it was for some other cultivations? Give a clearer statement in support of it.

L30: We rephrase it

9. L36: Give an appropriate reference to the term ‘conventional cacao system.’.

L38: A reference was provided.

10. L38: How all of a sudden it became a productive system? As just now you termed it to be simplified system requiring agrochemical dependency, which again was not properly managed by smallholders due to high input costs.

We corrected it. 

11. L39–43: This is good but, statement made- “this tendency towards ..... high input costs” and in this section are discrete. 

L36-41: We rephrase it to better understanding. 

12. L32–34: Is it reasonable to use the term agroforest/ agroforestry for Cacao system? Because, the system undergoes intense modification in respect to economic priority of the cultivars. Whereas, any agroforest should mimic the tree composition with the native forest or secondary forest. I think ‘Conventional Cacao System’ or ‘Conventional Cacao Agro-ecosystem’ would be more appropriate.

L35-39: We change the term. 

13. L56–57: Add more recent reference on expected global market demand and price hike on Cacao.

L57-59: We changed the sentence and provided more recent information.

14. L63: Add specified objectives of the study.

L64-69: Specific objectives was added. 

15. L74–75: On what basis you are terming them as young, middle age, etc. Also elaborate why <8 years cacao cultivation was deselected?

L:88-91: We add an explanation with reference for this selection criteria. 

16. L76: If it is 15×1.5 ha then result is 22.5 not 22.2. 

L95: We corrected this in all document. 

17. It looks that your qudrats (sampling plots) covered an area of 1.5 ha and not 22.2/22.5 ha. Clearly state what was the total area under laid plot not the area that was covered under cultivation? Because, your sampling time i.e., November and December raised questions in covering so much big area under quadrat laying. The section ‘In the afore mentioned ... 22.2 ha sampled’ and ‘Data collection was ... sub-plot was recorded needed rigorous modification to make it clearer to the audience.

It was rephrased to clearer for the audience.

18. L84–85: Farmers are mostly aware of vernacular names not scientific name and hence modify the sentence as an appropriately.

L97-100: We were modifying the sentence.

19. L89: Give official identification address of ‘KUELAP-UNTRM’ within the parenthesis.

L102: Address was provided.

20. L91–92: If you write this way then give the division factor which was used in conversion of CBH to DBH.

L109: The conversion factor was provided.

21. L93: ‘Procesamiento y análisis de datos’ Change it to English.

L110: Done

22. L94–103: Give specific references to each of the analysis performed. For example, for richness, for Shannon, Simpson, Margalef, Chao-1 indices, Jaccard and Sorenson.

L114: A references was provided. 

23. L112: ‘In general, when visualizing the total data...’ Rephrase this portion.

L137-138: It was rephrased. 

24. L119: ‘Species richness ranged from 1-9 in the plots.’ Table 2 showed Species richness at different aged stands ranged between 9 and 17.

a. L119–120: Modify to – ‘Species richness was the lowest at middle aged (16–29 years) cacao plots.’ Are you observing only Cacao trees in the entire cultivation system? If not, then why you are mentioning middle aged cacao trees? It should be middle aged of the cultivation system.

L144-149: We were rewriting this for better understanding. 

25. L123–126: ‘Simpson's (1-D) and Shannon's .... middle-age AFS are the least diverse compared to young and old AFS.’ This should be in discussion. Cite results of the diversity indices here.

L147-149: This sentence was rewrite. 

26. L128: At L121 you have mentioned that total 16 species were observed in the study area; here you are mentioning 454 individuals were distributed among 17 species distributed; and table 3 representing 17 species.

L151: We corrected, effectively it was 17 species. 

27. Table 3: Give references below table which was used to delineate the species as exotic, native etc. or add a section at method how you have distinguished this?

L103-105: We added a section at methods (L:91-93)

28. Table 3: What is potential products? Add a section in the manuscript how it was enumerated and how important it is in phytosociological studies?

L103-105: We added a sentence explaining how we enumerate the potential products.

29. L137–144: ‘The IVI reveals that .... with a low density of individuals.’ This is not proper result writing. You should give the Dr, Fr, Dor, IVI, etc. values representing the valuable species. This section is neither complete result nor complete discussion.

L59: we rewrite and added IVI values for the most important species to improve this sentence. 

30. How you have calculated IVI? Add a section in the method. Because it looks absolute value, and you are giving percent symbol. Better to use RIVI, this will give more clear expression of observed species.

L117-122: We added a section in “materials and methods” to explain how IVI was calculated. 

31. Figure 2: What are Joven, Madura and Adulta? If these are vernacular then give explanation of the term in figure caption. 

We corrected the language for figure.

32. L153–154: ‘there was a total of 8.23 m*ha2-1, 9.96 ..... and old cocoa AFS respectively.’ Should be written as- ‘Total basal area under young, middle-aged and old cacao AFS was observed to be 8.23 m2 ha-1, 9.96 m2 ha-1 and 10.95 m2 ha-1 respectively.

L177-179: We take the recommendation.

33. ‘m*ha2-1’: Correct it to ‘m2 ha-1’ across the whole manuscript. The digit 2 should be at superscript of m expressing as squared meter and -1 should be superscripted at ha denoting per hectare.

Done. 

34. L169: ‘Of the total .... in the three age rang’ change it to – ‘Seven of the total species studied were observed to be common in all the three age categories.’

L191: We take the recommendation

35. The term ‘reported’ should be changed to observed, found, etc.

Done

36. Figure 4: Change the language of axis title to English.

Done.

37. Shannon diversity index (H0) is not the correct expression. It should be H´.

The expression was checked in hole document.

38. L185–191: ‘The Cacao ecological zone .... looks notably promising.’ This section is not relevant for your research. I suggest change these explanations with relevant texts.

We had decided delete this paragraph.

39. At the very beginning of discussion avoid citing and linking with earlier reports. Start discussion with some productive phrases relevant to your findings.

We rephrasing and reorder this section to improve it. 

40. L200–201: A well-planned shade trees influences positive impact on cacao yield, improved net income together with various ecosystem services subject to implementation of proper management strategies (References).

L231-233: We take the recommendation and made the changes.

41. L213: ‘Côte d'Ivoire’, change it to English, if not possible give description within parenthesis.

L206: Changed

42. L222: that additionally provides other benefits (name some other benefits in this parenthesis) together with shade.

L229-230: We added this other benefits

43. L224–225: Micro-parcelling (give brief description of micro-parcelling) was also a common phenomena observed in the study area because of the presence relatively small cultivable plots (0.5–2 ha).

L241-243: We take this recommendation and rewrite the sentence.

44. L230–231: ‘since the density ... higher density of cacao plants.’ This explanation is not required.

We deleted it. 

45. L235¬–236: ‘which in young AFS .... so they have a lower density.’ It is a obvious fact so, this portion is not required.

We deleted it. 

46. L239–240: Jaccard’s and Sorenson’s are similarity indices. They evaluate similarity not dissimilarity. You can evaluate inverse Jaccard or inverse Sorenson for dissimilarity estimation. In such instances you have to explain how the dissimilarity or inverse similarity was evaluated in the methodology.

L125: We added a reference in the “Materials and Methods section”

---

## [Decision Letter · Decision Letter 1]

31 Aug 2022

PONE-D-22-03463R1Tree diversity in agroforestry systems of native fine-aroma cacao, Amazonas, PeruPLOS ONE

Dear Dr. Goñas,

Thank you for submitting your manuscript to PLOS ONE. After careful consideration, we feel that it has merit but does not fully meet PLOS ONE’s publication criteria as it currently stands. Therefore, we invite you to submit a revised version of the manuscript that addresses the points raised during the review process.

We look forward to receiving your revised manuscript.

Kind regards,

Arun Jyoti Nath

Academic Editor

PLOS ONE

Journal Requirements:

Additional Editor Comments:

 Please revise

Reviewers' comments:

Reviewer's Responses to Questions

**Comments to the Author**

1. If the authors have adequately addressed your comments raised in a previous round of review and you feel that this manuscript is now acceptable for publication, you may indicate that here to bypass the “Comments to the Author” section, enter your conflict of interest statement in the “Confidential to Editor” section, and submit your "Accept" recommendation.

Reviewer #1: (No Response)

2. Is the manuscript technically sound, and do the data support the conclusions?

Reviewer #1: Yes

3. Has the statistical analysis been performed appropriately and rigorously? 

Reviewer #1: Yes

4. Have the authors made all data underlying the findings in their manuscript fully available?

Reviewer #1: Yes

5. Is the manuscript presented in an intelligible fashion and written in standard English?

Reviewer #1: Yes

6. Review Comments to the Author

Reviewer #1: The manuscript has now improved and has achieved the level to be accepted for publication until the following minor points are addressed:

1. L21–24: Your presented results at L17–21 could not justify the conclusion made. Interpretation of results in the line of conclusion with similar discussion or modification of conclusion in the line of results whichever is feasible for you is suggested.

2. L25: Fruiting species or fruit bearing species

3. L38: Instead you write, Cacao is a socio-economically viable system....

4. L32–37: Remove this portion: However, farmers .... high input costs; because this is contradictory and it may lead reader to misinterpret conventional cacao and cacao AFS.

5. L92: Tree species were identified at the field itself ...

6. L111–112: Reference number 22 (IVI): Why you are citing so much old reference. Our data treatment and representation has improvised to a greater extent since then. You may check following some references for your IVI:

1) https://doi.org/10.1016/j.tfp.2020.100027 (Check IVI calculation and the supplementary table)

2) https://doi.org/10.1007/s11356-022-20329-4 (Check IVI calculation and the supplementary table)

7. Table 4: No need to put percent symbol against the values given for Dr, Fr, Dor. Because you have already cited in the column heading that the values are in percent. Check the above reference for representing IVI calculation and value representation.

8. Figure 2: m2 ha-1; here ‘2’ of m2 shoud be superscripted and ‘-1’ of ha-1 should be superscripted.

9. L170: m2 ha-1

7. PLOS authors have the option to publish the peer review history of their article (what does this mean?). If published, this will include your full peer review and any attached files.

Reviewer #1: **Yes: **Panna Chandra Nath

---

## [Author Response · Author response to Decision Letter 1]

8 Sep 2022

Dear editor and reviewers, We thank you all for the constructive comments provided. We believe that addressing every single one of them has improved significantly the manuscript. The response for specific comments are in the attached file labeled as "Response letter".

The authors

---

## [Editor Report · Decision Letter 2]

27 Sep 2022

Tree diversity in agroforestry systems of native fine-aroma cacao, Amazonas, Peru

PONE-D-22-03463R2

Dear Dr. Goñas,

We’re pleased to inform you that your manuscript has been judged scientifically suitable for publication and will be formally accepted for publication once it meets all outstanding technical requirements.

Kind regards,

Arun Jyoti Nath

Academic Editor

PLOS ONE
---

## [Editor Report · Acceptance letter]

3 Oct 2022

PONE-D-22-03463R2 

Tree diversity in agroforestry systems of native fine-aroma cacao, Amazonas, Peru 

Dear Dr. Goñas:

I'm pleased to inform you that your manuscript has been deemed suitable for publication in PLOS ONE. Congratulations! Your manuscript is now with our production department. 

Kind regards, 

on behalf of

Dr. Arun Jyoti Nath 

Academic Editor

PLOS ONE